# SIM: Intra-Group Member Differentiation via Social Interaction Modeling for Group Re-identification

## Abstract

Group Re-identification (G-ReID) focuses on associating group images that contain the same members across different camera views. The key challenge is that identity and position differentiation in group topology structure changes are difficult to capture. Drawing on principles from social psychology, we observe that the core members are more likely to remain in the group under different camera views with smaller position changes, while peripheral members are more likely to have significant position changes or even fade out of the group. To this end, we propose a novel social interaction modeling (SIM) method, which treats each group as a social interaction field to explore more authentic and robust group features through dealing with the member differentiation: identity and position differentiation. Our method constructs the social interaction calculation module (SICM) to capture the member differentiation in the fields, and implements identity differentiation and position differentiation by the social prior attention mechanism (SPAM) and social layout variation module (SLVM), respectively. Extensive experiments on three available datasets show that the proposed method SIM is effective, and outperforms all previous state-of-the-art methods, surpassing the baseline on Rank1/mAP by up to 8.6%/9.6% on DukeGroup, 3.7%/2.7% on RoadGroup and 2.5%/2.9% on CSG. The code will be available on Github.

## 1 Introduction

Group re-identification (G-ReID) aims to correctly associate group images that contain the same members captured by different cameras with non-overlapping views. It is increasingly important in the security field. G-ReID typically deals with groups of 2-6 members, and images belonging to the same group class should contain at least 60% same members. G-ReID is a more crucial and challenging task than person re-id because people usually have group and social attributes, indicating that people prefer to move in groups. Therefore, G-ReID needs to deal with the group topological structure changes: member and layout variation. Specifically, member variation means the members could leave the group, and layout variation means the positions may change under different cameras.

Although previous works (Zhang et al., 2022; 2024a; 2025) relied on uncertainty modeling to address the challenge of group topological structure changes, their performance was not satisfactory. The shortcomings are mainly due to the following two reasons: 1) The previous attention mechanism conducts undifferentiated learning for all intra-group member tokens, lacking specific focus. 2) The previous layout modeling method employs undifferentiated random affine transformations for the entire group, resulting in many ineffective layouts. Both shortcomings arise because existing methods ignore differentiation among intra-group members. As shown in Fig 1, the learned group features are undifferentiated for both member tokens and group layout, which are represented as pure pink triangles, leading to large intra-class distance.

By introducing the social psychology principles, we found that the core members are more likely to remain in the group with smaller position changes, and peripheral members are more likely to have significant position changes or even fade out of the group, which we term member differentiation. The identity differentiation and position differentiation mean the likelihood of appearing in other group images and the extent of position changes varies among intra-group members, respectively.

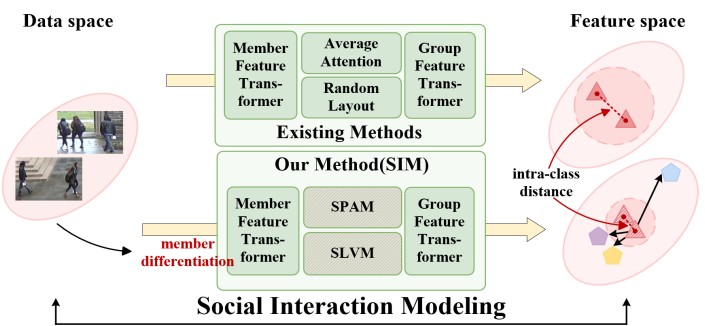 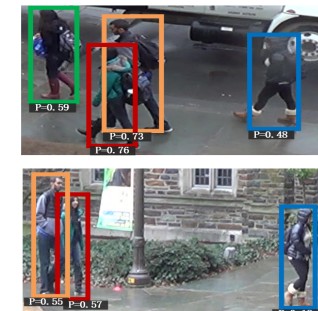

Figure 1: Existing methods versus social interaction modeling. Pure triangles denote group features extracted without member differentiation, while texture triangles indicate those extracted with differentiation. The pentagon represents the different member tokens that make up the group features. The dotted circle lines represent the boundaries within the class. The features extracted by our SIM have smaller intra-class distance due to consider the member differentiation.

Figure 2: Member differentiation. Two images belong to the same group, and $P$ represent the average interaction probability. B-boxes of same color represent the same member.

(Zhou et al., 2019) quantifies the formula between the interaction probability and three features of the social interaction field: distance, orientation, and pose-openness. The core members refer to the members with higher interaction probabilities relatively (e.g. members that be surrounded), while peripheral members refer to the members with lower interaction probabilities (e.g., unrelated pedestrians). As shown in Fig 2, members in red and orange b-box are with higher average interaction probability, remain their core position in the next group images, and members in blue and green b-box with lower probability, one has significant position changes, another left the group.

In this paper, we propose a novel social interaction modeling (SIM) method, which treats each group image as a social interaction field to mine more authentic and robust group features. We construct the social interaction calculation module (SICM) calculating interaction probabilities to capture the member differentiation in the fields, which is inspired by (Zhou et al., 2019). We further design the social prior attention mechanism (SPAM) and the social layout variation module (SLVM) to accomplish identity differentiation and position differentiation by utilizing these probabilities, respectively. As shown in 1, the group features learned by SIM are represented as texture pink triangles, which has smaller intra-class distance. Modeling and training this differentiation can obtain smaller intra-class distance and more authentic and robust group feature representations.

Specifically, the proposed SICM defines a normalized variable $\hat{p}$ to reflect the member differentiation. Each group image has a specific $\hat{p}$ and the accuracy of the $\hat{p}$-value is mainly affected by distance. SICM extracts social interaction features of members in the group image, which contain distance $d$, orientation $\theta$, openness $o$, which are extracted from annotations and group images.

SPAM aims to achieve identity differentiation by accounting for the varying likelihoods that individual members appear across different group images. SPAM adjusts the weight of attention for different members during group feature learning. To this end, a new attention weight allocation mechanism is designed to achieve core member mining and enhance group feature learning.

SLVM aims to address position differentiation. Due to the extent of position variation varies from intra-group members, SLVM models more realistic dynamic layout variations. Thus, a learnable position variation matrix $\triangle D$ is employed. While retaining a certain degree of freedom through weighted fusion with random affine vectors, SLVM accomplishes a new layout modeling strategy to conduct more realistic layout modeling and explore potential layout changes.

Our main contributions are summarized as follows:

- We first introduce the social psychology principles into G-ReID task, and propose the SIM method, which treats each group image as a social interaction field.

- We construct SICM to capture the member differentiation in the fields, and achieve identity and position differentiation by the proposed SPAM and SLVM to explore more authentic and robust group features.

## 2 RELATED WORK

**Person Re-identification.** Person re-identification (ReID) aims to associate individual pedestrians across non-overlapping views in camera networks. In recent years, numerous deep learning-based methods (Meng et al., 2021; Yan et al., 2020; Rao & Miao, 2023; Wang et al., 2024; Zhang et al., 2024b; Peng et al., 2023; Guo et al., 2024) have made significant progress in this field, including extracting more discriminative features and designing more suitable metrics. For instance, AGW (Ye et al., 2021) introduced a weighted regularized triplet metric learning method, while (Yang et al., 2025) Cheb-GR leverages Chebyshev-guided adaptive neighbor selection to enable efficient, training-free graph re-ranking for person Re-ID. However, person ReID methods primarily focus on individual pedestrians, overlooking the more intricate group-level interactions and layout dynamics that are pivotal for GReID.

**Group Re-identification.** Compared to ReID, research on G-ReID remains relatively scarce, with only a few pioneering works attempting to address this task. Early approaches (Zheng et al., 2011; Cai et al., 2010; Ristani et al., 2016; Lisanti et al., 2017) treated entire images as model inputs and directly extracted group features. Since these methods relied on handcrafted features and considered background information, their performance was unsatisfactory. Subsequently, CNN-based works MACG (Yan et al., 2020) proposed a multi-level attention contextual graph model to leverage visual context information among group members. In recent years, Vision Transformer-based architectures have gained popularity. UMSOT (Zhang et al., 2022; 2024a) introduced a second-order Transformer architecture to construct group features, incorporating uncertainty modeling of group member number and position. PBSOT (Zhang et al., 2025) proposed a parallel branches-based transformer with layout-guided occlusion mitigation, enhances robustness by strengthening the sampling of overlapping parts and fusing global features with local features. But existing methods take the entire-group perspective and overlook intra-group member differentiation.

**Social Interaction.** (Bolotta & Dumas, 2022) identified social interaction as a key area for future AI research in 2022, revealing that certain visual primitive features of social behavior discovered by cognitive psychologists enhance computer vision systems' ability to recognize interactions. (Chen et al., 2025) augments transformer-based pedestrian trajectory prediction by concurrently extracting and interacting subject-neighbor intentions across the entire observation period with a perception-masked attention mechanism. But these works rely only on pairwise distance and ignore the effects of orientation and pose openness. For example, two people standing close but back-to-back would not be considered a social group. SIFM (Zhou et al., 2019) found that closer interpersonal distances, more direct interpersonal angles and more open avatar postures led to a higher probability of a group being judged as interactive, and quantifies the formula between the interaction probability and social interaction features. However, they have not applied social-psychology principles to G-ReID, nor are they directly suitable.

## 3 METHOD

In this section, we first introduce how the SICM extracts social features and calculates interaction probabilities to capture member differentiation, and then describe how the proposed SPAM and SLVM implement identity differentiation and position differentiation. Fig 3 illustrates the method pipeline. For the $k$-th group, $x_k$ and $y_k^g$ are the group image and id, respectively, and $b_k$ and $y_k^p$ are the bounding box annotation and member id for each member in $x_k$, respectively.

### 3.1 SOCIAL INTERACTION CALCULATION MODULE (SICM)

In this paper, SICM aims to capture the member differentiation by calculating normalized interaction probabilities $\hat{p}_i$ in social interaction fields. The key issue is to extract social interaction features and calculate interaction probabilities of each group image.

For the $k$-th group image, we treat it as a social interaction field $S_k = \left\{ S_{ij}^k \right\}$, where member $i$ and $j$

have their subfield $S_{ij}^k$. A binary variable $z_{ij} \in \{0, 1\}$, following a Bernoulli distribution, determines whether pedestrians $i$ and $j$ are in the same social interaction field:

$$p\left(z_{ij} = 1 \mid S_{ij}^k\right) = \delta\left(S_{ij}^k\right). \tag{1}$$

To calculate the interaction probability $p_{ij}$ between member $i$ and $j$, we now extract social interaction features: distance, orientation and openness. Specifically, we utilize b-boxes $b_{ki}, b_{kj}$ to calculate distance $d_{ij}$. The distance between the $i$-th and $j$-th members based on their bboxes:

$$d_{ij} = \frac{1}{\gamma} \left\| \psi(b_{ki}^{mid}, b_{kj}^{mid}) \right\|, \tag{2}$$

where $b_{ki}^{mid}$ and $b_{kj}^{mid}$ denote the bottom midpoints of b-boxes, and $\gamma$ is a scaling factor. $\psi(x_1, x_2) = \phi(x_1) - \phi(x_2)$ is constructed to compensate for field-of-view discrepancies, where $\phi$ denotes a perspective transformation function (Zhang, 2021).

Then, we utilize multiple frameworks merging such as AlphaPose (Fang et al., 2022), Mediapipe (Lugaresi et al., 2019), HigherHRNet (Cheng et al., 2020) to extract skeletal keypoints $q_i$ from image of cropped member $x_{ki}$ to compute relative orientations $\theta_{ij}$ and define pose-openness $o_i, o_j$. We compute the relative angle:

$$\theta_{ij} = arccos \frac{v_i^\perp \cdot d_{ij}}{\left\| v_i^\perp \right\| \cdot \left\| d_{ij} \right\|}, \tag{3}$$

where $q_i^{ls}$ and $q_i^{rs}$ are the left and right shoulder keypoints of the $i$-th pedestrian, respectively. $v_i = q_i^{rs} - q_i^{ls}$ is shoulder vector of the $i$-th member, $v_i^\perp$ represents the orientation vector.

The pose-openness degree $o_i$ is defined as:

$$o_i = \begin{cases} 1, & \text{if } \zeta_{ub} > \zeta_1, \\ -1, & \text{if } q_i^{ws} \times q_i^{bd} > 0, \\ 0, & \text{otherwise.} \end{cases} \tag{4}$$

where $\zeta_{ub} = \left\langle q_i^{up}, q_i^{bd} \right\rangle$ is the angle between the upper arm $q_i^{up}$ and body $q_i^{bd}$, $\zeta_1$ is a threshold set to $45°$. $q_i^{ws}$ is forearm, and $\times$ is the cross product denotes vector outer product. Indicates that $o_i$ equal to 1 when upper arm is spread out, and $o_i$ equal to $-1$ when forearm tightens inward towards the body. Skeleton extraction accuracy on our three datasets is all above 97% (see Appendix E for detailed numbers and quantitative analysis under extraction failures).

Now we have social interaction features. Then we calculate the $p_{ij}$. The subfield intensity $S_{ij}^k$ and interaction probability satisfy the formula:

$$\begin{cases} S_{ij}^k = f(d_{ij}, \theta_{ij}, o_i, o_j) \cdot \delta\left(S_{ij}^k\right) \\ p_{ij} = 1 - exp(-S_{ij}^k/\lambda)^b \end{cases} \tag{5}$$

where function $f$ is a symmetric function for $i$ and $j$. The parameters $\lambda$, $b$ and function $f$ are all given in (Zhou et al., 2019). $P = \{p_{ij}\}_{i=1}^N$ is a symmetric matrix, because $S_{ij}^k = S_{ji}^k$. The average interaction probability $\bar{p}_i$, and normalized average interaction probability $\hat{p}_i$ can be described as:

$$\begin{cases} \bar{p}_i = \frac{1}{N-1} \sum_{j=1, j\neq i}^N p_{ij} \\ \hat{p}_i = \bar{p}_i / \sum_{i=1}^N \bar{p}_i = \frac{1}{N-1} \sum_{j=1, j\neq i}^N p_{ij} / \frac{1}{N} \sum_{i=1}^N \sum_{j=1, j\neq i}^N p_{ij} \end{cases} \tag{6}$$

where $N$ is the group size. For the $k$-th group $\hat{p}_k^g = \{\hat{p}_i\}_{i=1}^{N_k} \in \mathbb{R}^{N_k}$.

The control experiment for the independent effect of SICM and the feature ablation study in SICM are presented in Appendix F and Appendix H, respectively.

### 3.2 SOCIAL PRIOR ATTENTION MECHANISM (SPAM)

SPAM aims to achieve core member mining and enhance group feature learning by adjusting attention weights of group feature transformer, and accomplishes identity differentiation. The core

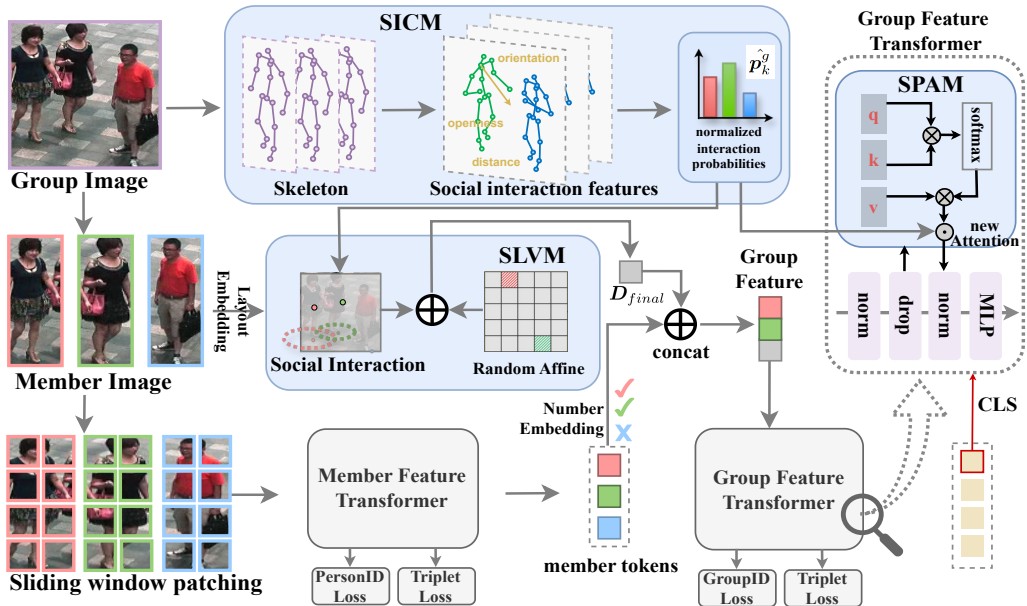

Figure 3: The pipeline of the proposed SIM. SICM is used to capture member differentiation with $\hat{p}_k^g$, SLVM is employed to model discriminative layout variation among members, and SPAM focuses on the tokens of core members. Group features are formed by concatenating member tokens and layout, serving as input to the Group Feature Transformer.

members are more likely to appear in other group images and their member IDs are more representative of the group ID. To this end, a new attention weight allocation mechanism SPAM is designed, which assigns them greater attention to enhance the tokens learning in group feature transformer while peripheral member receive less.

The $i$-th member's token $t_i^p$ of $k$-th group are extracted by ViT, $t_i^p = ViT(x_{ki})$, group feature is concatenated from members token, $t_k^g = [t_1^p, t_2^p, ..., t_{N_k}^p]$. The input of group vision transformer (GViT) are $X = [t_m^g, t_{m+1}^g, ..., t_{m+B-1}^g] \in \mathbb{R}^{B \times N \times C}$, where $B$ denotes the batch size, $N$ is the number of group members, $C$ represents the feature dimension, $p$ and $g$ represents person and group, $m$ due to current batch. The query $(Q)$, key $(K)$, value $(V)$ are obtained through linear transformations: $Q = XW_Q, K = XW_K, V = XW_V$, where $W_Q, W_K, W_V \in \mathbb{R}^{C \times d}$ are learnable parameters, and $d$ is the attention head dimension. The original attention weights are calculated as:

$$A_{raw} = (Q, K, V) = softmax\left(\frac{QK^T}{\sqrt{d}}\right)V. \tag{7}$$

The new attention is described as:

$$A_{social} = A_{raw} \odot \hat{\boldsymbol{P}}^g, \tag{8}$$

where $\hat{\boldsymbol{P}}^g = [\hat{\boldsymbol{P}}_1^g, \hat{\boldsymbol{P}}_2^g, ..., \hat{\boldsymbol{P}}_d^g] \in \mathbb{R}^{B \times C \times d}$, and $\hat{\boldsymbol{P}}_d^g = [\hat{\boldsymbol{p}}_m^g, \hat{\boldsymbol{p}}_{m+1}^g, ..., \hat{\boldsymbol{p}}_{m+B_d-1}^g] \in \mathbb{R}^{B \times C}$, and $\odot$ represents Hadamard product. The updated features $Z$ are generated through linear projection: $Z = A_{social}VW_O$, where $W_O \in \mathbb{R}^{d \times C}$ is a learnable projection matrix. By integrating $\hat{\boldsymbol{P}}^g$ and dimension alignment, SPAM optimizes attention weights, accomplishes identity differentiation.

## 3.3 SOCIAL LAYOUT VARIATION MODULE (SLVM)

SLVM aims to address position differentiation, accomplishes a new layout modeling strategy to conduct more realistic layout modeling and explore potential layout changes $D_{final}$. It constructs a learnable position variation matrix $\triangle D$, which restricts layout variation ranges for core members and expands it for peripheral members. This mechanism simulates realistic sociological layout variations that incorporate position differentiation.

Original layout coordinates of group image $D_{ori} = [(x_1, y_1), (x_2, y_2), ..., (x_N, y_N)] \in \mathbb{R}^{N \times 2}$ represents the center coordinates of each member in the image, $(x_i, y_i)$ represents the position of $i$-th member.

The existing methods treat the group layout as an entirety and apply random affine transformation, which fails to consider positional discrimination. The transformed layout after random affine transformation is given by: $D_{random} = RD_{ori} + b$, where $R \in \mathbb{R}^{2 \times 2}$ is random affine matrix, $b \in \mathbb{R}^2$ is translation vector.

For the position variation of $i$-th member of the group, our layout modeling can be described as:

$$\triangle d_i = \alpha \left( \sum_{j \neq i} \hat{p}_j(d_j - d_i) \right) + (1 - \alpha)r_i, \tag{9}$$

where $d_i$ and $d_j$ are the central position of $i$-th and $j$-th member's bbox, and $\sum_{i \neq j} \hat{p}_i(d_i - d_j)$ represents position differentiation. Hyperparameter $\alpha \in [0, 1]$ is a balancing coefficient that weights prior differentiation knowledge against data augmentation. $r_j$ is a random perturbation vector, and can maintain a certain degree of positional freedom for members.

The offset $\triangle d_j$ comprises two components: (1) a structure-aware offset, driven by social interaction weights, that preserves spatial proximity among core members; and (2) a random perturbation that introduces layout diversity.

Specifically, $(d_i - d_j)$ represents distance in real, $\hat{p}_i$ is normalized average interaction probability, which reflect member differentiation. Therefore, members with higher interaction probability with the $i$-th member can better maintain the distance between them and are accompanied by smaller positional variation. Meanwhile, members with lower interaction probability with the $i$-th member cannot maintain the distance between them and will generate greater layout variation.

The final layout is computed as:
$$D_{final} = D_{ori} + \triangle D, \tag{10}$$

where $\triangle D \in \mathbb{R}^{N \times 2}$ is the offset matrix for all intra-group members. The updated features $Z$ are fused with the layout information $D_{final}$ and Transformer encoder generates the group feature representation.

### 3.4 Loss function

Our feature is trained with person identity and triplet loss function.

$$\mathcal{L}_{ID} = -\frac{1}{P} \sum_{j=1}^{P} \sum_{i=1}^{C} y_{ji} log(\widehat{y}_{ji}), \tag{11}$$

where $P$ represents the total member number of the current batch, $C$ represents the total member classes, the indicator function $y_{ji} = 1(j = i)$ equals to 1 when the $j$-th member belongs to the $i$-th class, and $\widehat{y}_{ji}$ is the prediction of network about the $j$-th member belongs to the $i$-th class.

$$\mathcal{L}_{Tri} = \frac{1}{P} \sum_{i=1}^{P} max(d(f_i, f_i^+) - d(f_i, f_i^-) + h, 0), \tag{12}$$

where $d(\cdot, \cdot)$ represents the distance function between two features such as the Euclidean distance, $f_i / f_i^+ / f_i^-$ represent the anchor/hard positive/hard negative feature in the current batch, and $h$ is the hyper-parameter of margin. $\mathcal{L}_p$ is person loss $\mathcal{L}_p = \mathcal{L}_{ID} + \mathcal{L}_{Tri}$.

The loss function $\mathcal{L}_g$ of a second-order token (Zhang et al., 2024a) is also composed of the group identity and triplet loss, which is similar to the $\mathcal{L}_{ID}$ and $\mathcal{L}_{Tri}$. Overall, the total loss function is defined as:
$$\mathcal{L} = \mathcal{L}_p + \mathcal{L}_g. \tag{13}$$

Table 1: The proposed method is compared with state-of-the-art approaches on CSG, RoadGroup, and DukeGroup datasets. The comparative methods are divided into two categories: hand-crafted and deep learning-based methods. The best and second-best results are highlighted in bold and underlined, respectively. Reported metrics include Rank-1, Rank-5, Rank-10 and mAP (%).

| Method | Publication | CSG | | | | RoadGroup | | | | DukeGroup | | | |
|---|---|---|---|---|---|---|---|---|---|---|---|---|---|
| | | Rank1 | Rank5 | Rank10 | mAP | Rank1 | Rank5 | Rank10 | mAP | Rank1 | Rank5 | Rank10 | mAP |
| CRRRO-BRO | BMVC 2009 | 10.4 | 25.8 | 37.5 | - | 17.8 | 34.6 | 48.1 | - | 9.9 | 26.1 | 40.2 | - |
| Covariance | ICPR 2010 | 16.5 | 34.1 | 47.9 | - | 38.0 | 61.0 | 73.1 | - | 21.3 | 43.6 | 60.4 | - |
| BSC-CM | ICIP 2016 | 24.6 | 38.5 | 55.1 | - | 58.6 | 80.6 | 87.4 | - | 23.1 | 44.3 | 56.4 | - |
| PREF | ICCV 2017 | 19.2 | 36.4 | 51.8 | - | 43.0 | 68.7 | 77.9 | - | 30.6 | 55.3 | 67.0 | - |
| LIMI | MM 2018 | - | - | - | - | 72.3 | 90.6 | 94.1 | - | 47.4 | 68.1 | 77.3 | - |
| DotGNN | MM 2019 | - | - | - | - | 74.1 | 90.1 | 92.6 | - | 53.4 | 72.7 | 80.7 | - |
| DotSNN | TCSVT 2019 | - | - | - | - | 84.0 | 95.1 | 96.3 | - | - | - | - | - |
| GCGNN | TMM 2020 | - | - | - | - | 81.7 | 94.3 | 96.5 | - | 53.6 | 77.0 | 91.4 | - |
| SVIGR | Neucom 2020 | - | - | - | - | 87.8 | 92.7 | - | 89.2 | - | - | - | - |
| MGR | TCYB 2021 | 57.8 | 71.6 | 76.5 | - | 80.2 | 93.8 | 96.3 | - | 48.4 | 75.2 | 89.9 | - |
| MACG | TPAMI 2023 | 63.2 | 75.4 | 79.7 | - | 84.5 | 95.0 | 96.9 | - | 57.4 | 79.0 | 90.3 | - |
| SOT | AAAI 2022 | 91.7 | 96.5 | 97.6 | 90.7 | 86.4 | 96.3 | 98.8 | 91.3 | 72.7 | 88.6 | 93.2 | 78.9 |
| UMSOT | IJCV 2024 | 93.6 | 97.3 | 98.3 | 92.6 | 88.9 | 95.1 | **98.8** | 91.7 | 74.4 | 89.4 | 93.9 | 79.4 |
| PBSOT | ESWA 2025 | 94.5 | 97.1 | 97.9 | 93.9 | 91.3 | **96.3** | 98.7 | 93.3 | 82.7 | 92.6 | 95.1 | 88.1 |
| Ours | - | **96.1** | **98.3** | **99.1** | **95.5** | **92.6** | **96.3** | 97.5 | **94.4** | **83.0** | **96.6** | **98.9** | **89.0** |

# 4 EXPERIMENTS

## 4.1 DATASETS

The proposed SIM is evaluated on three G-ReID datasets: DukeGroup, RoadGroup (Lin et al., 2019), and CSG (Yan et al., 2020). The DukeGroup dataset contains 354 images with 177 group classes. The RoadGroup dataset contains 324 images with 162 group classes. Following the protocol in (Yan et al., 2020), the training and test sets of DukeGroup and RoadGroup are randomly divided equally. The CSG dataset contains 3,839 images with 1,558 group classes, where 859/699 groups are allocated for training/testing. According to (Yan et al., 2020), test images are sequentially selected as probes, while all remaining images serve as the gallery. Additionally, CSG includes 5K extra group images in the gallery as distractors. For fair comparison, no additional datasets are used during training on any G-ReID dataset. Evaluation metrics include Rank-1, Rank-5, Rank-10 cumulative matching characteristics (CMC) and mean average precision (mAP). Experiments of distractors on dataset CSG are given in Appendix I.

## 4.2 DETAILS

Our baseline is UMSOT. Experiments were conducted on an NVIDIA RTX 4080 GPU using Py-Torch. Our model uses GViT (Zhang et al., 2024a) as backbone, pretrained on ImageNet (Deng et al., 2009). For input group images, we crop all member images using given bounding boxes and resize them to 256×128. During training, the random seed is fixed to 42, with random horizontal flipping (p=0.5) and random erasing applied. Each mini-batch samples 16 group identities, with 4 images selected per identity. We use SGD (Bottou, 2012) as optimizer. Training terminates after 400 epochs. A cosine annealing learning rate schedule is employed: initial rate 2e-3, minimum rate 1.6e-4. The learning rate for inter-member modules is multiplied by 0.1. Weight decay is set to 1e-4. Online hard mining is used for triplet loss (Hermans et al., 2017). During testing, no data augmentation or re-ranking is applied. Features are compared using Euclidean distance. Unless otherwise specified, all experiments are conducted on the DukeGroup dataset.

## 4.3 PERFORMANCE

We compare SIM with existing methods on three available G-ReID datasets to demonstrate its superiority. We categorize existing methods into two groups: handcrafted G-ReID methods and deep learning-based G-ReID methods, From the performance perspective, SIM is recognized as the state-of-the-art method among existing approaches. Two conclusions can be drawn from Table 1:

First, our SIM achieves strong performance on the CSG, RoadGroup, and DukeGroup datasets, surpassing baseline in both Rank-1 and mAP metrics. On the CSG dataset, SIM achieves 96.1%/95.5% (Rank-1/mAP), outperforming baseline by 2.5%/2.9 in Rank-1/mAP. On the RoadGroup dataset, SIM achieves 92.6%/94.4% (Rank-1/mAP), outperforming baseline by 3.7%/2.7% in Rank-1/mAP. On the DukeGroup dataset, SIM achieves 83.0%/89.0% (Rank-1/mAP), outperforming baseline by

Table 2: Ablation Experiments on the Social Interaction Module (SIM) (%).

| SPAM | SLVM | CSG | | RoadGroup | | DukeGroup | |
|------|------|-------|-------|-------|-------|-------|-------|
| | | Rank1 | mAP | Rank1 | mAP | Rank1 | mAP |
| | | 93.57 | 92.62 | 88.89 | 91.73 | 74.42 | 79.40 |
| ✓ | | **96.11** | 95.28 | 91.36 | 93.75 | 80.68 | 87.48 |
| | ✓ | 95.59 | 95.22 | 89.94 | 92.23 | 78.41 | 85.88 |
| ✓ | ✓ | 96.06 | **95.49** | **92.59** | **94.35** | **82.95** | **89.02** |

8.6%/9.6% in Rank-1/mAP. These results demonstrate that SIM delivers consistent performance gains across all datasets, confirming the effectiveness of member differentiation and yielding significant improvements over the baseline. Second, the performance of existing method remains unsatisfactory due to: Existing methods take the entire-group perspective and do not consider member differentiation in group topology structure changes. Unlike these methods, SIM constructs SICM to capture member differentiation in social interaction fields. The proposed SPAM achieves core member mining and enhance group feature learning, while SLVM accomplishes a new layout modeling strategy to conduct more realistic layout modeling and explore potential layout changes, making SIM shortening the intra-class distance, enhancing the robustness of group features. Inference speed & memory on DukeGroup: SIM 0.838 s / 6225 M and UMSOT 0.806 s / 6207 M (+4 % time, +0.3 % memory), while accuracy improves from 74.4 % to 83.0 % R-1 and 79.4 % to 89.0 % mAP.

## 4.4 ABLATION STUDY

**Effect of SPAM and SLVM.** The ablation experiments primarily demonstrate the impact of the SPAM and SLVM modules on social interaction modeling. We mainly analyze the results on the DukeGroup dataset, with similar conclusions observed on the other two datasets. As shown in Table 2, two conclusions can be drawn: First, each module individually improves performance when used separately. Compared to the baseline, SPAM increases Rank-1/mAP by +6.26%/+8.08%, while SLVM increases Rank-1/mAP by +3.99%/+6.48%. This indicates that these two modules respectively learn the identity and position differentiation, making SIM more discriminative. Second, when both modules are used together, the performance gains reach +8.53%/+9.62% in Rank-1/mAP, exceeding the sum of individual improvements. This demonstrates that SPAM and SLVM are two complementary aspects for mining group differentiated features. Using both SPAM and SLVM simultaneously enables the exploration of more authentic and robust group features.

## 4.5 PARAMETER ANALYSIS

**Influence of $\alpha$.** The hyperparameter $\alpha$ controls how SLVM learns potential group layouts by determining the ratio between the social interaction layout matrix and the random affine matrix during training in SIM. When $\alpha$ is set too large, SLVM over-emphasizes layout variations of peripheral members, leading to overfitting and performance degradation. When $\alpha$ is set too small, SLVM fails to adequately explore potential layout features of peripheral members, resulting in insufficient model generalization and performance decline. As shown in Figure 4, we conducted separate hyperparameter experiments on CSG, RoadGroup and DukeGroup datasets to determine optimal values, since each dataset exhibits different social interaction fields distributions.

## 4.6 VISUALIZATION

**Retrieval visualization.** Figure 5 presents the top-3 retrieval visualizations comparing baseline UMSOT and our proposed SIM. The advantages of SIM are primarily demonstrated in two aspects: 1) Achieve core member mining and enhance group feature learning, achieve identity differentiated learning. UMSOT tends to retrieve gallery images with higher overall similarity to the query. When processing groups with less distinctive members (Rows 1, 2, and 4), SIM effectively focuses on core members of the group. 2) Conduct more realistic layout modeling and explore potential layout changes, achieve position differentiated learning. UMSOT does not emphasize layout variations. For groups in Rows 3 and 5, SIM better captures the topological changes of the group structure.

**Feature visualization.** Figure 6 illustrates the group feature visualization of our best eight classes in the training set when both UMSOT and our proposed SIM converge, and each group class con-

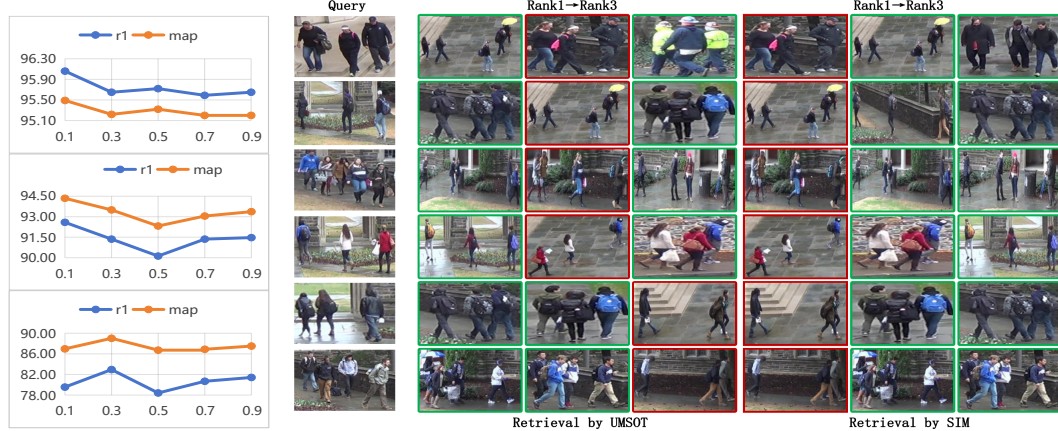

Figure 4: Parameter $\alpha$ on CSG, RoadGroup and DukeGroup.

Figure 5: Top-3 Retrieval Visualization of UMSOT and SIM. Red/green bounding boxes indicate correct/incorrect matches, respectively. In the DukeGroup dataset, each query has only one correct match in the gallery.

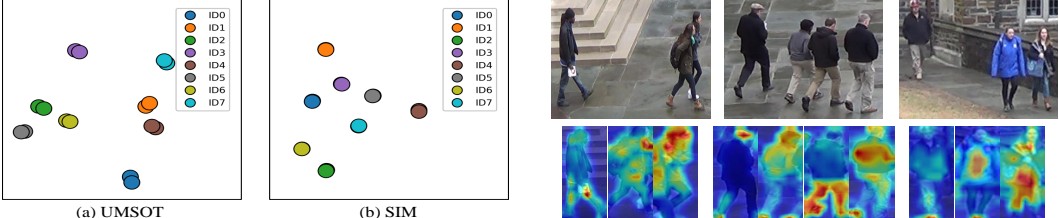

Figure 6: The feature visualization of the whole training set through t-SNE (Van der Maaten & Hinton, 2008). Each color represents a group class.

Figure 7: Heatmap visualization via Grad-CAM is applied to dataset images. Pedestrians with higher social interaction probability exhibit larger heatmap areas.

tains only two images. Due to UMSOT's approach regards groups as entire distributions during the feature learning process, the intra-class distance is large. In contrast, our social interaction modeling utilizes the member differentiation in learning group features, demonstrating: excellent intra-class consistency and strong inter-class differentiation.

**Heatmap.** We optimize group feature learning by adjusting the weights in the group feature Transformer attention mechanism–SPAM, and visualize individual member features combined with group weight using Grad-CAM (Selvaraju et al., 2017) class activation heatmaps as shown in Figure 7. The results show that members with higher social interaction probability exhibit larger and more concentrated heatmap regions, indicating that the learning of group features is optimized and backpropagates to affect individual representations.

## 5 CONCLUSION

In this paper, we focus on the member differentiation in group topology structure changes in G-ReID, which contains identity and position differentiation. To solve this, we propose a novel social interaction modeling method, which treats group as a social interaction field. First, our method constructs SICM to capture the member differentiation in fields, and accomplish identity differentiation and position differentiation by SPAM and SLVM, respectively, shorten the intra-class distance. Second, SPAM design a new attention weight allocation mechanism, to mine core member and enhance group feature learning. SLVM achieve a new layout modeling strategy to conduct more realistic layout modeling and explore potential layout changes. Finally, our proposed social interaction modeling achieves state-of-the-art performance across multiple benchmark datasets, outperforming all existing methods. Limitations are detailed in Appendix A.

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

# A   LIMITATIONS.

Although the proposed SIM method has achieved promising results in methods and effects, it still has some undeniable limitations. First, to reflect the members differentiation, the calculation of interaction probability requires skeleton extraction for orientation, which is restricted by current basic techniques. Although we employ multiple pose estimation models, the accuracy of skeleton extraction remains imperfect. Interaction probability is mainly determined by the distance, although the errors caused by the orientation and openness are very small, they still exist. Second, owing to restrictions on public release and incomplete annotations of additional datasets, all experiments were conducted on the three most widely used public benchmarks.

# B   CONTRIBUTION TO OPTIMIZING THE ATTENTION MATRIX.

## B.1   THEOREM 1 (MEMBER-DIFFERENTIATION LOWER BOUND)

let $C = \{j \mid p_{ij} \geq \tau\}$ and $E = \{j \mid p_{ij} < \tau\}$; then

$$\mathbb{E}_{j \in C}[\boldsymbol{A}_{\text{social}}]_j - \mathbb{E}_{j \in E}[\boldsymbol{A}_{\text{social}}]_j \geq \tau \big(\mathbb{E}_{j \in C}[\boldsymbol{A}_{\text{raw}}]_j - \mathbb{E}_{j \in E}[\boldsymbol{A}_{\text{raw}}]_j\big). \qquad (14)$$

Hence the Hadamard product **amplifies** the attention gap between core and periphery **linearly with** $\tau$.

# C   THE HADAMARD PRODUCT ENHANCES PERFORMANCE.

## C.1   THEOREM 2 (CONVERGENCE)

for cross-entropy loss $\mathcal{L} = -\log \text{softmax}(\boldsymbol{A}_{\text{raw}} \odot \boldsymbol{P})$,

$$\frac{\partial \mathcal{L}}{\partial p_{ij}} = -(\delta_{ii} - \delta_{ij})[\boldsymbol{A}_{\text{raw}}]_{ij} \qquad (15)$$

High-$p_{ij}$ pairs obtain **larger gradients**, speeding up learning of core-member features and suppressing peripheral noise, which improves **both accuracy and convergence speed**.

# D   MEMBER INTERPRETATION

members with **larger average** $\hat{p}_i$ change position less and stay longer in the group; SPAM therefore assigns them **higher attention**, while low-$\hat{p}_i$ members receive **less focus**.

## D.1   LEMMA 1 (GRADIENT MONOTONICITY)

$$\frac{\partial p_{ij}}{\partial d_{ij}} = -\frac{2d_{ij}}{\lambda} \exp\Big(-\frac{\|\boldsymbol{s}_{ij}\|_2^2}{\lambda}\Big) < 0, \quad \frac{\partial^2 p_{ij}}{\partial d_{ij}^2} > 0.$$

Distance influence **decays convexly**, guaranteeing that only nearby members significantly affect $p_{ij}$.

# E   QUANTIFY THE IMPACT OF SKELETON KEYPOINT EXTRACTION FAILS ON PERFORMANCE

The three datasets used in our experiments—CSG, RoadGroup, and DukeGroup—have relatively high image quality. Therefore, we simulated the practical value on low-quality skeleton data by using different skeleton extraction strategies with varying success rates.

The experimental results using AlphaPose and Mediapipe as skeleton-extraction strategies are presented in Table 3 and Table 4.

It can be seen that even on low-quality skeleton data, the performance remains significantly higher than the baseline.

Table 3: Experimental results using AlphaPose as skeleton-extraction strategy (%).

| Dataset | AlphaPose | | | | |
|---|---|---|---|---|---|
| | Extraction accuracy | Rank1 | Rank5 | Rank10 | mAP |
| CSG | 99.04 | 96.06 | 98.26 | 99.07 | 95.49 |
| RoadGroup | 100.00 | 92.59 | 97.53 | 98.77 | 94.30 |
| DukeGroup | 97.97 | 82.95 | 96.59 | 98.86 | 89.02 |

Table 4: Experimental results using Mediapipe as skeleton-extraction strategy (%).

| Dataset | Mediapipe | | | | |
|---|---|---|---|---|---|
| | Extraction accuracy | Rank1 | Rank5 | Rank10 | mAP |
| CSG | 79.59 | 95.48 | 97.68 | 98.61 | 95.05 |
| RoadGroup | 78.04 | 91.36 | 96.30 | 97.53 | 93.75 |
| DukeGroup | 51.48 | 80.68 | 94.32 | 97.73 | 86.71 |

## F   A CONTROL EXPERIMENT FOR THE INDEPENDENT EFFECT OF SICM

To determine the independent contribution of SICM, we replaced the full features with a distance-only weighting scheme Table 5:

**Performance drops** only marginally, confirming that orientation and pose-openness provide a positive but small gain, and that **distance is the dominant feature** in SICM.

## G   THE UPDATE MECHANISM OF THE LEARNABLE POSITION VARIATION MATRIX $\Delta \mathbf{D}$

**Complete Group Feature.** Due to space limits, Section 3.2 only states "group feature is concatenated from member tokens, $\mathbf{t}_k^g = [\mathbf{t}_1^p, \ldots, \mathbf{t}_{N_k}^p]$". The **full group feature** explicitly concatenates member tokens with the final layout matrix:

$$\mathbf{t}_k^g = [\mathbf{t}_1^p, \ldots, \mathbf{t}_{N_k}^p, \ \mathbf{D}_{\text{final}}], \qquad \text{where} \quad \mathbf{D}_{\text{final}} = \mathbf{D}_{\text{ori}} + \Delta \mathbf{D}. \tag{16}$$

**Forward Process.** Final layout coordinates are updated as

$$\mathbf{D}_{\text{final}} = \mathbf{D}_{\text{ori}} + \Delta \mathbf{D}, \qquad \Delta \mathbf{D} \in \mathbb{R}^{N \times 2}. \tag{17}$$

Row-wise perturbation (for member $i$):

$$\Delta \mathbf{d}_i = \alpha \sum_{j \neq i} p_j (\mathbf{d}_j - \mathbf{d}_i) + (1 - \alpha) \mathbf{r}_i, \qquad \alpha \in [0, 1], \ \mathbf{r}_i \sim \mathcal{N}(\mathbf{0}, \sigma^2 \mathbf{I}), \tag{18}$$

with normalised interaction probabilities $p_j$.

**Back-Propagation Derivation.** Let $\mathcal{L}$ denote the final loss (triplet + ID). By the chain rule,

$$\frac{\partial \mathcal{L}}{\partial \Delta \mathbf{D}} = \frac{\partial \mathcal{L}}{\partial \mathbf{D}_{\text{final}}}. \tag{19}$$

Gradients of (18) are

$$\frac{\partial \Delta \mathbf{d}_i}{\partial \alpha} = \sum_{j \neq i} p_j (\mathbf{d}_j - \mathbf{d}_i) - \mathbf{r}_i, \tag{20}$$

$$\frac{\partial \Delta \mathbf{d}_i}{\partial \mathbf{d}_j} = \alpha p_j \mathbf{I} \quad (j \neq i), \tag{21}$$

$$\frac{\partial \Delta \mathbf{d}_i}{\partial \mathbf{d}_i} = -\alpha \sum_{j \neq i} p_j \mathbf{I}. \tag{22}$$

Equations (21)-(22) show that neighbours with **high** $p_j$ contribute more gradient to $\Delta \mathbf{d}_i$; thus the displacement of **core members is suppressed** while that of **peripheral members is amplified**.

Table 5: Independent effect of SICM (%). The weights calculated by distance-only and SICM (distance, orientation and openness)

| Dataset | Distance-only | | | | SICM | | | |
|---------|-------|-------|--------|------|-------|-------|--------|------|
| | Rank1 | Rank5 | Rank10 | mAP | Rank1 | Rank5 | Rank10 | mAP |
| CSG | 95.2 | 98.1 | 98.8 | 94.8 | 96.1 | 98.3 | 99.1 | 95.5 |
| RoadGroup | 90.1 | 96.3 | 97.5 | 92.9 | 92.6 | 96.3 | 97.5 | 94.4 |
| DukeGroup | 79.6 | 96.6 | 98.9 | 87.1 | 83.0 | 96.6 | 98.9 | 89.0 |

Table 6: Feature ablation study in SICM on RoadGroup (%).

| Distance | Orientation | Openness | Rank1 | Rank5 | Rank10 | mAP |
|----------|-------------|----------|-------|-------|--------|------|
| ✓ | ✓ | ✓ | 92.59 | 97.53 | 98.77 | 94.30 |
| ✓ | ✓ | – | 91.36 | 96.30 | 96.30 | 93.68 |
| ✓ | – | ✓ | 90.12 | 97.53 | 97.53 | 93.14 |
| ✓ | – | – | 90.12 | 96.30 | 97.53 | 92.86 |

## H  THE FEATURE ABLATION STUDY IN SICM

We ablated the features of SICM on the RoadGroup dataset while keeping the original formula f unchanged: Table 6

**The results show that** removing either cue causes only a minor drop, confirming that distance is the dominant factor while orientation and pose-openness provide small but positive contributions.

## I  EXPERIMENTS OF DISTRACTORS ON DATASET CSG

This is the experiments on the relationship between the number of distractors samples, model inference time, and Rank-1 accuracy. As mentioned in our paper, "CSG includes 5K extra group images in the gallery as distractors." Therefore, we conduct experiments with 1K, 3K, and 5K distractors to analyze how SIM's retrieval efficiency changes as the number of distractor samples increases. Table 7

Table 7: Experiments on the relationship between the number of distractor samples, model inference time (s/batch), and Rank-1 accuracy (%).

| Distractors | Rank1 | Rank5 | Rank10 | mAP | Inference time (s/batch) |
|-------------|-------|-------|--------|------|--------------------------|
| 5K | 96.06 | 98.26 | 99.07 | 95.49 | 0.046 |
| 3K | 96.52 | 98.43 | 98.96 | 95.90 | 0.042 |
| 1K | 97.45 | 99.01 | 99.30 | 96.91 | 0.030 |

