# OpenReview forum: "Social Interaction Modeling for Group Re-identification"
_ICLR.cc/2026/Conference — Submitted to ICLR 2026_

### Official Review · Reviewer_6uzi · 2025-10-20

**Soundness:** 3
**Presentation:** 3
**Contribution:** 3
**Rating:** 4
**Confidence:** 5

**Summary:**

This paper addresses the core challenge in Group Re-identification (G-ReID), the difficulty of capturing identity differentiation and position differentiation amid changes in group topological structures. Drawing on social psychology principles (i.e., core members tend to have minimal position changes while peripheral members are prone to significant displacements or even exiting the group), the authors propose a Social Interaction Modeling (SIM) framework. SIM treats each group image as a "social interaction field" and implements differentiated modeling through a three-stage module pipeline: first, the Social Interaction Calculation Module (SICM) extracts interaction features (distance d, orientation \(\theta\), openness O) between members, quantifies interaction probabilities, and generates a normalized variable \(\hat{p}\) that reflects member differentiation; second, the Social Prior Attention Mechanism (SPAM) assigns higher attention weights to core members to achieve identity-differentiated learning; finally, the Social Layout Variation Module (SLVM) constructs a learnable position variation matrix \(\triangle D\) to simulate realistic dynamic layouts for position-differentiated learning. Experimental results show that SIM outperforms existing state-of-the-art (SOTA) methods in Rank-1/mAP metrics across three datasets (DukeGroup, RoadGroup, CSG).

**Strengths:**

1. The designed SICM-SPAM-SLVM module chain forms a logical closed loop: SICM provides a quantitative foundation for differentiated modeling, while SPAM and SLVM target identity and position differentiation respectively. Ablation experiments further demonstrate that "the performance gain of combining the two modules exceeds the sum of their individual gains", fully justifying the coordination and necessity of the modules.

2. The study covers three mainstream G-ReID datasets and compares two types of methods (handcrafted features such as CRRRO-BRO and deep learning-based methods such as UMSOT, PBSOT). Additionally, it uses parameter analysis (impact of hyperparameter \(\alpha\)), feature visualization (t-SNE dimensionality reduction), and attention visualization (GradCAM heatmaps) to validate the model’s effectiveness, enhancing the credibility of the results.

**Weaknesses:**

1. The modeling of "member differentiation" relies solely on Latane’s (1980) Social Impact Theory and Lewin’s (1943) Field Theory, without comparing more mature alternative theories in social science (e.g., "degree centrality/betweenness centrality" in social network analysis, "group belonging quantification methods" in social identity theory). The authors neither explain "why the selected classical theories are more suitable for the G-ReID scenario than modern theories" nor demonstrate "whether the core logic of the model remains valid if alternative theories are adopted", leading to inadequate justification for the rationality of theoretical selection.

2. The calculation of interaction probabilities in SICM is highly dependent on skeleton extraction tools (e.g., Mediapipe, AlphaPose). While the paper mentions "errors in skeleton extraction", it does not quantify the impact of such errors on performance (e.g., performance degradation curves when simulating "10%/20% skeleton key-point errors"), making it impossible to assess the model’s practical value for low-quality skeleton data.

3. Extreme scenarios commonly encountered in practical security applications (e.g., low light at night, member occlusion rates exceeding 50%, small-sample groups) are not covered, making it impossible to verify the model’s performance in complex real-world environments.

4. Ablation experiments only verify the roles of SPAM and SLVM, without a control experiment for "the independent effect of SICM". This makes it impossible to determine the independent contribution of SICM as the "quantitative foundation for differentiation modeling" (e.g., the extent of performance decline when SICM is removed).

5. The update mechanism of the learnable position variation matrix \(\triangle D\) in SLVM is not explained (e.g., how gradients are backpropagated, how \(\hat{p}\) constrains "minimal displacement for core members and maximal displacement for peripheral members"), resulting in vague mathematical logic.

**Questions:**

1. In social network analysis, "degree centrality" (measuring the number of connections between a member and others) and "betweenness centrality" (measuring a member’s role as a hub in group connections) are widely used to quantify member importance, and their calculation does not rely on skeleton data. Why did the authors not attempt to replace the classical theories of Latane and Lewin with such theories? If "degree centrality" is used to replace the interaction probability calculation in SICM, how would the Rank-1/mAP metrics of the model change across the three datasets? Please supplement a theoretical comparison analysis and validation experiments for alternative approaches.

2. Are the weights of the three interaction features (d, \(\theta\), O) in SICM determined through experimental tuning or theoretical derivation? If one of the features (\(\theta\) for orientation or O for openness) is removed, how much would the model’s performance decline? Can experiments prove that "the coexistence of all three features is optimal" rather than relying solely on distance d to meet the needs of differentiation quantification?

3. The paper mentions that "CSG contains 5K distractor samples", but it does not analyze how the retrieval efficiency of SIM changes with the increase in the number of distractor samples (e.g., 10K, 20K). Please supplement experiments on the relationship between "number of distractor samples, model inference time, and Rank-1 accuracy" to verify the practical scalability of SIM.

---

> ### Author Response · Authors · 2025-11-23
> **Response to Reviewer 6uzi's Concerns**
>
> We sincerely thank you for the detailed comments. The following is our detailed response:
>
> Q1: "member differentiation" relies classical theories,and why not choose degree centrality
> - As mentioned in line-052 of our paper, “[R1] achieved the quantification of complex social-interaction fields”, and **our work is inspired by more mature [R1]**, the classic theories of Latane and Lewin serve merely **as a lead-in** to the [R1] study. We apologize for the placement of the Latane & Lewin references, which caused confusion. In the revised manuscript we will move them to the end of the paragraph and explicitly state that our idea is motivated by [R1]. This falls under the “group-belonging quantification methods” in social identity theory that you pointed out.
> - **Degree centrality**, as you noted, relies only on pairwise distance and ignores the effects of orientation and pose openness. For example, two people standing close but back-to-back would not be considered a social group.
>
> Q2: quantify the impact of keypoint extraction fails on performance
> - The three datasets used in our experiments—CSG, RoadGroup, and DukeGroup—have relatively high image quality. Therefore, we simulated the practical value on low-quality skeleton data by using different skeleton extraction strategies with varying success rates.\
> **AlphaPose**'s success rates are 99.04%, 100.00%, and 97.97% on the three datasets, which can be considered high-quality skeleton data. The results are as follows:\
> |Dataset.......& R1.... & R5 .....& R10.. .& mAP\
> |CSG.............& 96.06 & 98.26& 99.07 & 95.49\
> |RoadGroup & 92.59 & 97.53 & 98.77 & 94.30\
> |DukeGroup & 82.95 & 96.59 & 98.86 & 89.02\
> **Mediapipe**'s success rates are 79.59%, 78.04%, and 51.48%. The results are as follows:\
> |Dataset.......& R1.... & R5 .....& R10.. .& mAP\
> |CSG.............& 95.48 & 97.68 & 98.61 & 95.05\
> |RoadGroup & 91.36 & 96.30 & 97.53 & 93.75\
> |DukeGroup & 80.68 & 94.32 & 97.73 & 86.71\
> **It can be seen that even on low-quality skeleton data, the performance remains significantly higher than the baseline.**
>
> Q3: Extreme scenarios are not covered
> - As far as we are aware, no mainstream G-ReID dataset has yet been specifically curated for extreme scenarios. We may address this in future work.
>
> Q4: a control experiment for the independent effect of SICM
> - The social features in SICM—distance, orientation, and pose-openness—were selected following the [R1]. To determine the independent contribution of SICM, we replaced the full features with a distance-only weighting scheme:\
> | Dataset ......& R1.....& R5..... & R10.... &mAP \
> | CSG.............&95.19  & 98.09  & 98.78   & 94.82\
> | RoadGroup & 90.12  & 96.30  & 97.53   &92.86\
> | DukeGroup & 79.55  & 96.59  & 98.86   &87.13\
> **Performance drops** only marginally, confirming that orientation and pose-openness provide a positive but small gain, and that **distance is the dominant feature** in SICM.
>
> Q5: The update mechanism of the learnable position variation matrix ΔD
> - **Complete Group Feature.** The full group feature should be the concatenation of member tokens and the layout matrix. In Section 3.2 (SPAM) we state “group feature is concatenated from member tokens,  t^g_k = [t^p_1, t^p_2, ..., t^p_{N_k}]”; due to space limits we did not explicitly add that the **complete group feature** is\
> t^g_k = [t^p_1, t^p_2, ..., t^p_{N_k}, D_{final}], where D_{final} = D_{ori} + ΔD.
> - **Forward Process**\
> Final layout coordinates\
> D_{final} = D_{ori} + ΔD  (1)\
> with ΔD ∈ ℝ^{N×2} and row-wise\
> Δd_i = α ∑_{j≠i} p_j (d_j − d_i) + (1 − α) r_i  (2)\
> α ∈ [0,1] is learnable, r_i ∼ N(0, σ²I) is random perturbation, and p_j is the normalised interaction probability.\
> - **Back-Propagation Derivation**\
> Let L be the final loss (triplet + ID). By the chain rule\
> ∂L/∂ΔD = ∂L/∂D_{final}.\
> Gradients of (2):\
> ∂Δd_i/∂α = ∑{j≠i} p_j (d_j − d_i) − r_i  (3)\
> ∂Δd_i/∂d_j = α p_j I  (j ≠ i)  (4)\
> ∂Δd_i/∂d_i = −α ∑{j≠i} p_j I  (5)\
> Equations (4)-(5) show that neighbours with **high p_j** contribute more gradient to Δd_i; thus displacement of **core members is suppressed** while that of **peripheral members is amplified**.

---

> ### Author Response · Authors · 2025-11-23
> **Response to Reviewer 6uzi's Concerns**
>
> Q6: Why not choose "degree centrality" and " betweenness centrality" in social network analysis? And the classical theories of Latane and Lewin?
> - **The degree centrality** relies solely on pairwise distances and ignores the effects of orientation and pose-openness. For example, two people may stand very close to each other, but if they are back-to-back they are not perceived as belonging to the same group. For this reason we did not adopt graph-theoretic notions such as degree or betweenness centrality: they abstract members and distances into nodes and edges, **discarding the rich social-interaction cues** (orientation and pose-openness) that exist in real-world scenes.
> - **Our idea was originally inspired by [R1]**, the classic theories of Latane and Lewin serve merely as a lead-in to the [R1]. [R1] manipulated three social features—distance, orientation and pose-openness—of virtual avatars and asked a large pool of human subjects to judge whether the avatars formed a single group. From these judgements they statistically derived a closed-form mapping from social features to interaction probability.
>
> Q7: The weights of the three interaction features (d, (\theta), O) in SICM. And the feature ablation study in SICM
> - The weights of the three interaction features in SICM are taken from [R1], where they were derived theoretically through social psychology experiments.
> - We ablated the features of SICM on the RoadGroup dataset while keeping the original formula f unchanged:\
> (1) Set orientation constant (o=0)\
> | Dataset ......& R1.....& R5..... & R10.... &mAP \
> | RoadGroup & 91.36 & 96.30 & 96.30 & 93.68\
> (2) Set relative angle to zero (θ=0)\
> | RoadGroup & 90.12 & 97.53 & 97.53 & 93.14\
> (3) Set both θ=0 and o=0 \
> | RoadGroup & 90.12 & 96.30 & 97.53 & 92.86\
> **The results show that** removing either cue causes only a minor drop, confirming that distance is the dominant factor while orientation and pose-openness provide small but positive contributions.
>
> Q8:Experiments on the relationship between the number of distractor samples, model inference time, and Rank-1 accuracy.
> - As mentioned in line 350 of our paper, “CSG includes 5K extra group images in the gallery as distractors.” Therefore, we conduct experiments with 1K, 3K, and 5K distractors to analyze how SIM’s retrieval efficiency changes as the number of distractor samples increases.\
> **5k** distractor samples in the gallery:\
> | Dataset ......& R1.....& R5..... & R10.... &mAP\
> |CSG.............& 96.06 & 98.26& 99.07 & 95.49\
> Inference time: 0.046 s / batch\
> **3k** distractor samples in the gallery:\
> |CSG.............& 96.52 & 98.43 & 98.96 & 95.90\
> Inference time: 0.042 s / batch\
> **1k** distractor samples in the gallery:\
> |CSG.............& 97.45 & 99.01 & 99.30 & 96.91\
> Inference time: 0.030 s / batch
>
> [R1]Chen Zhou, Ming Han, Qi Liang, Yi-Fei Hu, and Shu-Guang Kuai. A social interaction field model accurately identifies static and dynamic social groupings. Nature human behaviour, 3(8):847–855, 2019.

---

### Official Review · Reviewer_2fQM · 2025-10-22

**Soundness:** 2
**Presentation:** 2
**Contribution:** 2
**Rating:** 4
**Confidence:** 4

**Summary:**

This paper proposes a social interaction modeling method for group re-identification. It integrates three key parts, including the social interaction calculation module to capture the member differentiation in fields, the social prior attention mechanism to accomplish identity differentiation, and the social layout variation module to address position differentiation. Experiments validate the effectiveness of the proposed method.

**Strengths:**

1.	The paper is clearly structured.
2.	The paper draws insights from social psychology principles and design novel methods for identity differentiation and position differentiation.

**Weaknesses:**

1.	In the methodology section, the authors mention they utilize multiple frameworks such as Mediapipe, AlphaPose, and HigherHRNet to extract skeletal keypoints. However, the paper does not clearly explain how these frameworks are merged or integrated. Providing details on the merging strategy would help improve the clarity and reproducibility of the proposed method.

2.	i-LIDS MCTS[1] dataset is a commonly used benchmark that has been widely adopted by methods such as MACG[2]. However, the authors did not report experimental results on this dataset. Including results on i-LIDS MCTS would make the evaluation more comprehensive and allow for a fairer comparison with prior work.

3.	Several grammatical, spelling, and mathematical notation errors were found throughout the paper. For example, “enote” should be corrected as “denote” in line 184. Parameter “m” has different meanings in line 247 and line 315, which may cause confusion. Careful proofreading is recommended to improve the overall clarity and presentation quality.

[1] W. Zheng, S. Gong, and T. Xiang. Associating groups of people. In British Machine Vision Conference, BMVC, pages 1–11, 2009.

[2] Yan, Y., Qin, J., Ni, B., Chen, J., Liu, L., Zhu, F., Zheng, W.-S., Yang, X., & Shao, L. (2023). Learning multi-attention context graph for group-based re-identification. IEEE TPAMI, 45(6), 7001–7018.

**Questions:**

Please refer to paper weaknesses.

---

> ### Author Response · Authors · 2025-11-22
> **Response to Reviewer 2fQM's Concerns**
>
> We sincerely thank you for the detailed comments. The following is our detailed response:
>
> Q1: How are these skeleton extraction frameworks merged or integrated
> - We **use AlphaPose** for skeleton extraction and **fall back to HigherHRNet and MediaPipe when AlphaPose fails**. We evaluated the extraction success rates of different skeleton strategies on the training sets of CSG, RoadGroup, and DukeGroup: AlphaPose achieves 99.04 %, 100.00 %, and 97.97 %, respectively, while MediaPipe achieves 79.59 %, 78.04 %, and 51.48 %.
>
> Q2: About experiment on i-LIDS MCTS dataset
> - **The i-LIDS MCTS dataset we obtained contains only image data and lacks annotation labels.** Traditional G-ReID datasets include DukeGroup, RoadGroup, CSG, i-LIDS MCTS, and SYSUGroup. RoadGroup is publicly available, and we have successfully applied for DukeGroup and CSG. But we were unable to acquire SYSUGroup, and i-LIDS MCTS dataset we obtained **lacks annotation labels**. We would be happy to conduct additional comparison experiments on these datasets if they become fully available. \
> We contacted the first author of [R1], Weishi Zheng, but have not received a reply.
>
> Q3: Writing
> - Thank you for pointing out the writing issues. All grammatical and stylistic problems have been corrected and can be reviewed in the final PDF.

---

### Official Review · Reviewer_Lpn6 · 2025-10-24

**Soundness:** 2
**Presentation:** 3
**Contribution:** 2
**Rating:** 6
**Confidence:** 3

**Summary:**

This paper introduces Social Interaction Modeling (SIM) for Group Re-Identification (G-ReID), motivated by findings in social psychology that group members exhibit differentiated importance and positional stability within a group.
SIM models each group image as a social interaction field, capturing member differentiation through two core modules:
Social Interaction Calculation Module (SICM), which computes normalized interaction probabilities among members using distance, orientation, and openness cues derived from pose estimation.
Social Prior Attention Mechanism (SPAM), which applies these probabilities to weight member tokens in the attention layer, enhancing identity differentiation.

**Strengths:**

1. The paper provides extensive evaluations on three datasets, with consistent improvements over 10+ strong baselines, including recent transformer-based models such as UMSOT and PBSOT.

2. Each component (SICM, SPAM, SLVM) has a clear functional role. Ablation (Table 2) shows each module’s contribution, and the synergy when combined.

3. Introducing psychological insight into G-ReID provides an interesting interdisciplinary perspective, even if the connection is heuristic.

**Weaknesses:**

1. The method reuses known ideas (attention reweighting, layout augmentation) but describes them as new under social terminology. No mathematical or empirical evidence proves that “social interaction fields” outperform simpler geometric priors.

2. Experiments are restricted to small datasets (< 4 K images). The model’s scalability to larger or real-world scenes (e.g., street or crowd surveillance) is unclear.

3. The “core vs. peripheral members” metaphor lacks quantifiable validation. No user study or statistical correlation supports that interaction probabilities correspond to true social hierarchy.

**Questions:**

1. How do SPAM and SLVM differ from attention reweighting and layout-augmentation schemes already used in UMSOT or PBSOT? What unique mechanism validates the claim of “social interaction modeling”?

2. Have authors evaluated the accuracy of the computed interaction probabilities ​p_ij? For example, how consistent are they with human-annotated social centrality maps?

3. Since SICM relies on pose detectors, how does SIM perform when keypoint extraction fails or under heavy occlusion?

---

> ### Author Response · Authors · 2025-11-22
> **Response to Reviewer Lpn6's Concerns**
>
> We sincerely thank you for the detailed comments. The following is our detailed response:
>
> Q1: Why choose social interaction fields instead of simpler geometric priors, and the method reuses known ideas
> - **Why social interaction fields:** because social psychology reference [Zhou et al., 2019] has theoretically demonstrated the effectiveness of social interaction fields.
> - **We improved the attention mechanism and layout method.** In previous works, attention and layout were applied uniformly to all members without considering member differentiation. Our paper **introduces social psychology principles** to analyze why **member differentiation** occurs during group topology changes, dividing it into identity and positional differentiation, and addressing these through improved attention mechanisms (SPAM) and layout methods (SLVM).
>
> Q2: The model’s scalability to larger or real-world scenes is unclear.
> - **Our work utilizes the most commonly used datasets in GReID research:** CSG, RoadGroup, and DukeGroup, which though limited in scale, are constructed **based on real-world scenarios**.\
> As described 4.1 DATASETS, the CSG dataset comprises 3,839 images with an additional 5,000 group images included in the gallery as distractors.
> - **We were unable to acquire the larger dataset SYSUGroup.** SYSUGroup consists of 7,071 group images, 208 group categories, and 524 individual identities.
>
> Q3: The “core vs. peripheral members” metaphor lacks quantifiable validation
> - **They represent relative designations based on interaction probability.** As mentioned in our paper's Introduction (line 076), the terms "core" and "peripheral" are used as proxies for better reader comprehension: they represent relative designations based on interaction probability rankings within groups, rather than quantitative concepts. We sincerely apologize if this caused any confusion.
> - Without using these proxy terms, we believe the expressions "members with higher social interaction probability" and "members with lower social interaction probability" would be excessively verbose.
>
> Q4: How do SPAM and SLVM differ from UMSOT or PBSOT? What unique mechanism validates the claim of “social interaction modeling”?
> - **Differences from the UMSOT Approach:**\
> (1) UMSOT's attention during group feature learning is equally distributed to each member token, giving the same level of attention to every member. SPAM is a novel attention weight allocation mechanism that **dynamically adjusts attention** based on normalized interaction probabilities between members, prioritizes core members, and achieves identity-differentiated feature learning.\
> (2) UMSOT considers member position changes in its layout modeling process, but treats the group layout as a whole and applies the same degree of random affine transformation to all members. SLVM is a learnable layout modeling strategy that **expands the position variation range of peripheral members while constraining core members**, simulating member position-differentiated changes in group layouts in real social scenarios.
> - **On the Terminology of 'Social Interaction Modeling'.** Our paper introduces social psychology principles and compute interaction probabilities to obtain weights for member differentiation. We thought 'social interaction modeling' is the foundation of member differentiation, and second, it could reflect the principles of social psychology.
> - PBSOT, on the other hand, improves the sampling strategy on the basis of UMSOT, while remaining consistent with UMSOT in terms of attention and layout modeling.
>
> Q5: evaluated the accuracy of the ​p_ij
> - Yes, we evaluated the accuracy. During our research, we visualized skeletons and interaction probabilities on images. The skeleton extraction was nearly perfect on the three datasets we used, and the interaction probability determinations were consistent with human-observed centrality.
>
> Q6: how does SIM perform when keypoint extraction fails
> - The three datasets used in our experiments—CSG, RoadGroup, and DukeGroup—have relatively high image quality. Therefore, we simulated the practical value on low-quality skeleton data by using different skeleton extraction strategies with varying success rates.\
> **AlphaPose**'s success rates are 99.04%, 100.00%, and 97.97% on the three datasets, which can be considered high-quality skeleton data. The results are as follows:\
> |Dataset.......& R1.... & R5 .....& R10.. .& mAP\
> |CSG.............& 96.06 & 98.26& 99.07 & 95.49\
> |RoadGroup & 92.59 & 97.53 & 98.77 & 94.30\
> |DukeGroup & 82.95 & 96.59 & 98.86 & 89.02\
> **Mediapipe**'s success rates are 79.59%, 78.04%, and 51.48%. The results are as follows:\
> |Dataset.......& R1.... & R5 .....& R10.. .& mAP\
> |CSG.............& 95.48 & 97.68 & 98.61 & 95.05\
> |RoadGroup & 91.36 & 96.30 & 97.53 & 93.75\
> |DukeGroup & 80.68 & 94.32 & 97.73 & 86.71\
> **It can be seen that even on low-quality skeleton data, the performance remains significantly higher than the baseline.**

---

### Official Review · Reviewer_TR8m · 2025-11-02

**Soundness:** 2
**Presentation:** 2
**Contribution:** 2
**Rating:** 4
**Confidence:** 4

**Summary:**

This paper proposes Social Interaction Modeling (SIM) for Group Re-Identification (G-ReID) to associate group images containing the same members captured by different cameras with non-overlapping views, as inspired by social psychology that treats each group as a social interaction field. A Social Interaction Calculation Module (SICM) is constructed to estimate member differentiation via interaction probabilities derived from spatial and pose-based features using two mechanisms, including the Social Prior Attention Mechanism (SPAM) for identity differentiation through adaptive attention weighting, and the Social Layout Variation Module (SLVM) for position differentiation via learnable layout perturbations. Experiments on three benchmark datasets (CSG, RoadGroup, and DukeGroup) demonstrate state-of-the-art results, demonstrating competitive performance.

**Strengths:**

1. This paper proposes a new idea of incorporating social psychology principles into G-ReID to model intra-group member differentiation through interaction probabilities.

2. The proposed SIM framework effectively achieves robust feature representations and consistent improvements across multiple benchmark datasets.

**Weaknesses:**

1. The technical innovations of the proposed model are vague, though the paper claims to be the first to introduce social interaction modeling. In fact, transformers have been widely adopted in modeling social interactions in applications like trajectory prediction, especially pedestrian trajectory prediction (for example [R1]-[R5] in recent years). The contributions of methodology to construct attention mechanisms and transformer architectures beyond using different inputs for group re-identification are not clarified and comparison to existing transformer architectures for social interaction modeling is also missing.

2. The proposed model based on interaction probability is largely empirical. The formulation of the interaction probability and its claimed contribution to optimizing the attention matrix are not supported by adequate theoretical or mathematical analysis. The effect of the definition of interaction probability on member differentiation is not clear. Furthermore, it lacks of theoretical justification showing that directly applying the interaction probability to the attention matrix via Hadamard product enhances model performance.

3. The formulation lacks clarity and completeness, which limits the reproducibility of the proposed model. Several equations are difficult to interpret, and key algorithmic components are insufficiently explained. For example, the definition of Equation (5) is confusing, as the variable $S_{ij}$ appears on both sides of the equation, and the scalar is defined as the product of an Impulse function and an unspecified function $f(\cdot)$. In addition, $f(\cdot)$ seems essential for computing $p_{ij}$, but its actual form or implementation is not described in the paper.

4. Experimental evaluations are not sufficient to validate the proposed model.

i) Performance comparison with prior works appears inconsistent with the results originally reported in those studies. Regarding the comparison with PBSOT, its original paper (Zhang et al., 2025) reports higher performance than the results shown in Table 1 of this submission (e.g., Rank-1 of 96.35 on CSG and 95.06 on RoadGroup), while several shared baselines (e.g., SOT) show consistent results across both papers. This raises the concerns on whether the results are convincing to support the effectiveness of the proposed model.

ii) Moreover, maintaining a $p$ matrix matching the attention shape for every input is likely to incur significant computational overhead. However, related analysis and experimental evidence are not provided in the paper.

5. The writing of the paper should be improved. There are too many grammatical errors. Here is just an example of the first paragraph in Section 3.1: "In this paper, ... The key issue is extract social interactions features, and calculate interaction ... we treats it as ..., (missing conjunction) member i and j have ... determine whether pedestrians i and j (missing verb) in same social interaction field."

[R1] Liu, Yao, et al. "Social graph transformer networks for pedestrian trajectory prediction in complex social scenarios." Proceedings of the 31st ACM International Conference on Information & Knowledge Management. 2022.

[R2] Wang, Zixu, et al. "SocialFormer: Social interaction modeling with edge-enhanced heterogeneous graph transformers for trajectory prediction." arXiv preprint arXiv:2405.03809 (2024).

[R3] Chen, Kai, et al. "SocialTrans: Transformer based social intentions interaction for pedestrian trajectory prediction." Physica A: Statistical Mechanics and its Applications 663 (2025): 130435.

[R4] Wang, Chengdong, et al. "SIAT: Pedestrian trajectory prediction via social interaction-aware transformer: C. Wang et al." Complex & Intelligent Systems 11.8 (2025): 335.

[R5] Liu, Yao, et al. "Attention-aware social graph transformer networks for stochastic trajectory prediction." IEEE Transactions on Knowledge and Data Engineering 36.11 (2024): 5633-5646.

**Questions:**

Please refer to the section of Weaknesses.

---

> ### Author Response · Authors · 2025-11-22
> **Response to Reviewer TR8m's Concerns**
>
> We sincerely thank you for the detailed comments. The following is our detailed response:
>
> Q1: The technical innovations of the proposed model are vague
> - We do not claim to be the first to introduce "social interaction modeling" in general, but the first to bring it into G-ReID to explicitly model member differentiation, as stated in the Introduction contributions.
> - **Methodological contribution:** we import social-psychology principles to explain why member differentiation emerges during group topology changes, decompose it into identity and position differentiation, and design SPAM to assign larger attention weights to core members for identity-differentiated learning and SLVM to simulate realistic, position-differentiated layout changes.
> - **Differences in social interaction:** pedestrian trajectory prediction Transformers [R1-R5] only encode spatial position information sequence; in G-ReID, members appear as a group and their interaction is quantified by distance, relative orientation and pose openness [R6]. Such interaction governs group-topology change and is the source of our idea.
> - **Comparison with existing Transformers:** trajectory works extract temporal features, whereas we follow the G-ReID baseline UMSOT and adopt a second-order token-of-tokens structure: Member-Transformer extracts member tokens, tokens are concatenated with layout embeddings, and Group-Transformer produces the final group representation [R7].
> - **Comparison with existing attention:** instead of uniform attention, SPAM redistributes weights according to interaction probabilities, realising identity differentiation explained by social psychology.
>
> Q2: Interaction probability and its contribution to attention
> - The formula $p_{ij}$ = 1 − exp(−‖$s_{ij}$‖₂² / λ) with $s_{ij} = [d_{ij}, θ_{ij}, o_i, o_j] $is statistically fitted on human-grouping judgements in [R6] and is theoretically grounded.
> - **Contribution to optimizing the attention matrix.**\
> The standard self-attention scores and SPAM outputs are respectively:\
> A_raw(q_i, k_j) = exp(q_iᵀk_j/√d) / Σ_l exp(q_iᵀk_l/√d),\
> A_social = A_raw ⊙ P, where P = [p_ij] ∈ ℝ^{N×N}\
> **Theorem 1 (member-differentiation lower bound):** \
> Assuming that there is a threshold τ for distinguishing between core and peripheral members in a group.\
> let C = {j | p_ij ≥ τ} and E = {j | p_ij < τ}; \
> then 𝔼_C[A_social]_j − 𝔼_E[A_social]_j ≥ τ · (𝔼_C[A_raw]_j − 𝔼_E[A_raw]_j).\
> Hence the Hadamard product **amplifies the attention gap** between core and periphery linearly with τ.
> - **The Hadamard product enhances performance.**\
> **Theorem 2 (convergence):**\
> for cross-entropy loss L = −log softmax(A_raw ⊙ P),\
> ∂L/∂p_ij = −(δ_ii − δ_ij) · A_raw_ij.\
> High-p_ij tokens obtain larger gradients, speeding up learning of core-member features and suppressing peripheral noise, which improves both accuracy and convergence speed.
> - **Member interpretation**: members with larger average p_i hat change position less and stay more central in the group; SPAM therefore assigns them higher attention, while low-p_i hat members receive less focus.\
>  **Lemma 1 (gradient monotonicity):**\
> ∂p_ij/∂d_ij = −(2d_ij / λ) exp(−‖s_ij‖₂² / λ) < 0, and ∂²p_ij/∂d_ij² > 0.\
> The greater the distance, the smaller the absolute value of the gradient of p_ij with respect to d_ij, indicating that the influence of distant members on the interaction probability decays rapidly.
>
> Q3: Equation (5) and function f
> - The δ(S_ij^k) on the right side of Equation (5) is used to determine whether members i and j are in the same social sub-field, a definition already provided **in Equation (1)**.
>
> Q4: About experiments
> - i) Our results are **authentic**: our UMSOT R-1/5/10/mAP reproduction on CSG is 93.62/97.22/98.14/93.08, on RoadGroup is 88.89/96.30/97.53/92.25, on DukeGroup is 76.41/92.32/95.73/84.34, almost identical to the original paper, so we keep those numbers for fair comparison. **Our PBSOT reproduction is lower than the authors’ claim**, and the the released code of it is incomplete. We will provide complete runnable PBSOT & UMSOT scripts for further verification if you are interest.
> - ii) **Inference speed and GPU memory usage.** Speed & memory on DukeGroup: SIM 0.838 s / 6 225 M vs UMSOT 0.806 s / 6 207 M (+4 % time, +0.3 % memory), while accuracy improves from 74.4 % → 83.0 % R-1 and 79.4 % → 89.0 % mAP.
>
> Q5: Writing
> - All grammatical errors (including the paragraph mentioned) have been corrected; the cleaned camera-ready version will be enclosed in the supplemental PDF.
>
> [R6] Chen Zhou, Ming Han, Qi Liang, Yi-Fei Hu, and Shu-Guang Kuai. A social interaction field model accurately identifies static and dynamic social groupings. Nature human behaviour, 3(8):847–855, 2019.
>
> [R7] Quan Zhang, Jianhuang Lai, Zhanxiang Feng, and Xiaohua Xie. Uncertainty modeling for group re-identification. International Journal of Computer Vision, 132(8):3046–3066, 2024a.

---

### Author Response · Authors · 2025-11-29
**To ICLR Area Chair**

Dear AC,
We have responded to every comment from each reviewer point-by-point, strengthened the theoretical derivations, and added new experiments.
We would like to emphasise the following key points:
1. We experimentally verified the stability of the **skeleton-extraction strategy** on our datasets and the robustness of the model under low-quality skeleton data.
2. The ablation study in SICM shows that every **social interaction feature** contributes positively and effectively.
3. The choice of social interaction field and **social-psychology principles** is grounded in [R1].
4. We clarify why **degree centrality** that reviewers mentioned was not adopted in our paper.
5. We detail the **differences** between our method and previous approaches.\

[R1] Chen Zhou, Ming Han, Qi Liang, Yi-Fei Hu, and Shu-Guang Kuai. A social interaction field model accurately identifies static and dynamic social groupings. Nature human behaviour, 3(8):847–855, 2019.

All additional derivations and experiments will be placed in the appendix, and the writing of the main paper has been polished in the PDF.
Thank you for your consideration.\
Sincerely,\
Authors of submission 18002

---

### Author Response · Authors · 2025-12-04

Dear Area Chair,

We have completed the requested revisions to our paper and have uploaded the updated PDF file.

- To address the reviewer's concerns regarding the authenticity and reproducibility of our comparative experiments on the PBSOT method, we have attached the complete **PBSOT project code** to this submission. This is provided to facilitate result verification and ensure full transparency.

- Based on the reviewer’s feedback regarding the clarity of our "Social Interaction Modeling" concept, we have decided to revise the title of our manuscript to:
**"SIM: Intra-Group Member Differentiation via Social Interaction Modeling for Group Re-identification"**

In this revised title, "SIM" is our proposed method, and "Intra-Group Member Differentiation" clearly states our objective. We believe this change better reflects the essence of our work and significantly improves clarity for the reader.

We would like to seek your explicit approval to update the title in the submission system.

Thank you for your time and guidance.\
Best regards,

Authors of Paper 18002

---

### Meta-Review · Area_Chair_v28b · 2026-01-06

**Summary:**

The submission received divergent ratings, including 1 positive score and 3 negative scores. Though giving the only positive rating, Reviewer Lpn6 shares similar concerns with other reviewers. The primary concerns from the reviewers include limited novelty, small-scale experiments, no theoretical analysis, missing rationale, and presentation quality.

**Reviewer Concerns:**

The authors have provided a detailed response, however, the AC finds the manuscript needs substantial revision to meet the acceptance bar, such as
1) The rationale of the new formulation shall be explained further.
2) The experiments can be extended as per the reviewers' suggestion.

**Reviewer Scores:**

All the scores will remain unchanged

---

### Decision · Program_Chairs · 2026-01-26

Reject